# Epsilon Toxin from *Clostridium perfringens* Causes Inhibition of Potassium inward Rectifier (Kir) Channels in Oligodendrocytes

**DOI:** 10.3390/toxins12010036

**Published:** 2020-01-06

**Authors:** Jean Louis Bossu, Laetitia Wioland, Frédéric Doussau, Philippe Isope, Michel R. Popoff, Bernard Poulain

**Affiliations:** 1Institut des Neurosciences Cellulaires et Intégratives, (INCI)-CNRS, UPR 3212 Strasbourg, France; jlbossu@inci-cnrs.unistra.fr (J.L.B.); Laetitia.Wioland@univ-grenoble-alpes.fr (L.W.); doussau@unistra.fr (F.D.); philippe.isope@inci-cnrs.unistra.fr (P.I.); 2Institut Pasteur, Bactéries Anaérobies et Toxines, 28 rue du Docteur Roux, Paris 75724, France; popoff2m@gmail.com

**Keywords:** oligodendrocyte, epsilon toxin, multiple sclerosis, myelin, inward rectifying potassium channel, *Clostridium perfringens*

## Abstract

Epsilon toxin (ETX), produced by *Clostridium perfringens* types B and D, causes serious neurological disorders in animals. ETX can bind to the white matter of the brain and the oligodendrocytes, which are the cells forming the myelin sheath around neuron axons in the white matter of the central nervous system. After binding to oligodendrocytes, ETX causes demyelination in rat cerebellar slices. We further investigated the effects of ETX on cerebellar oligodendrocytes and found that ETX induced small transmembrane depolarization (by ~ +6.4 mV) in rat oligodendrocytes primary cultures. This was due to partial inhibition of the transmembrane inward rectifier potassium current (Kir). Of the two distinct types of Kir channel conductances (~25 pS and ~8.5 pS) recorded in rat oligodendrocytes, we found that ETX inhibited the large-conductance one. This inhibition did not require direct binding of ETX to a Kir channel. Most likely, the binding of ETX to its membrane receptor activates intracellular pathways that block the large conductance Kir channel activity in oligodendrocyte. Altogether, these findings and previous observations pinpoint oligodendrocytes as a major target for ETX. This supports the proposal that ETX might be a cause for Multiple Sclerosis, a disease characterized by myelin damage.

## 1. Introduction

*Clostridium perfringens* strains of types B and D produce epsilon toxin (ETX), a ~30 kD peptide with very high lethality (~400,000 mouse lethal doses/mg protein, intraperitoneally) [1,2]. ETX is category B among the agents of particular concern [1]. Its high lethality is due to its high neurotoxicity [3,4].

ETX is produced by *C. perfringens* in the lumen of the intestine and is disseminated throughout the body by the bloodstream. The toxin binds to the endothelial cells of the capillary vessels of the brain [5,6,7]. Using the tetraspan proteolipid Myelin and Lymphocyte Protein (MAL) as a receptor, it is then transcytosed by a caveol-dependent process [7]. Once in the brain parenchyma, ETX—at sub-nanomolar concentration—attacks mature oligodendrocytes, responsible for axonal myelination. Indeed, ETX binds to oligodendrocytes, myelin and white matter [6,8,9,10]. The binding of ETX to oligodendrocytes depends on the expression of MAL [11]. In organotypic cerebellar slices of rodents, ETX triggers marked demyelination of neuronal axons [9,10,12]. In vivo, sub-lethal doses of ETX induce demyelination [13] and alter nerve conduction [14]. It is important to note that patients suffering from multiple sclerosis, a disease characterized by impaired nerve conduction due to demyelination, have a much frequent immuno-reactivity against ETX than that determined in healthy control group (10% vs 1% [15]; 23% vs 8% [16]. Therefore, it has been proposed that ETX exposure may play a causal role in the first stage of this chronic disease [15,16].

The mechanisms underlying the effects of ETX on oligodendrocytes leading to demyelination are not yet clear. Studies conducted on oligodendrocytes in rats in primary culture have shown that ETX causes an increase in extracellular glutamate that is not due to permeabilization of the plasma membrane of oligodendrocytes [9]. Elevation of extracellular glutamate results in the activation of metabotropic type 1 glutamate receptors (mGluR1) and N-Methyl-D-Aspartate receptors (NMDA-R) in oligodendrocytes. In turn, this triggers intracellular Ca^2+^ signaling and demyelination [9]. Since the activity of glutamate glial membrane transporters is highly dependent on transmembrane potential (V_m_), it is possible that ETX modifies V_m_ without affecting the integrity of the plasma membrane.

The membrane inward rectifier potassium channels (Kir channels) play an important role in maintaining the Nernst equilibrium potential for K^+^ ions (EK^+^) and adjusting the resting potential of the membrane (RMP) near E_K_^+^ [17,18]. Maintaining the RMP at a hyperpolarized value is essential for membrane excitability, regulation of transmembrane transporters (including glutamate), signalling, differentiation, and in particular oligodendrocyte myelination of neuronal axons [17,18,19,20]. In mature oligodendrocytes, Kir channels are the main types of active channels present in the plasma membrane [17,18,21,22]. Their voltage-dependency comes from the voltage-dependent block of the open channel pore by intracellular divalent cations (such as Mg^2+^) and other molecules (such as polyamines) at V_m_ more positive than E_K_^+^ [18]. This gives them rectifying characteristics in allowing inward flux of K^+^ ions at the V_m_ more negative than E_K_^+^ while impeding outward K^+^ current at the depolarized V_m_ [18].

In this paper, experiments were sought to examine whether ETX modifies RMP or/and Kir membrane current and channels in mature oligodendrocytes from rat cerebellum. Using primary cultures, we demonstrate that the application of ETX induces a small transmembrane depolarization and inhibition of a fraction of Kir transmembrane current. Moreover, we found that one of the two main types of Kir channels present in oligodendrocytes is inhibited following the ETX application.

## 2. Results

### 2.1. ETX Causes Small Oligodendrocyte Membrane Potential Depolarization

Using the whole-cell clamp technique (current-clamp mode), we examined whether the extracellular application ETX modifies RMP of oligodendrocytes. In 15 experiments (15 petri-dishes, 1 oligodendrocyte recorded per dish), RMP was first determined for 5 min (baseline) and after ETX (10^−7^ M, final) was applied in the bath. After ETX, RMP was determined at the plateau of depolarisation (average V_m_ determined for 1s). The average RMP was −55.1 ± 2.3 mV before ETX (i.e., control period) and −49.9 ± 2.8 mV after (*n* = 15, *p* < 0.001 Wilcoxon Signed Rank Test). Of the 15 cells that were recorded, only 12 responded to ETX by a V_m_ change. The latter manifested as an onset of depolarization followed by a plateau (Figure 1). Cells were considered as non-responding to ETX when their V_m_ remained unchanged for a period of at least 7 min after the toxin was applied to bath (i.e., with V_m_ similar to baseline ±0.5 mV and without step). The delay between ETX application and onset of depolarization was determined on the 12 cells responding to ETX by determining the time interval between the application artifact and the starting point for the depolarization onset (i.e., the intersection between linear adjustments of the baseline level and depolarization onset). In these 12 cells, after ETX, depolarization started after an average delay of 53 ± 12 s and the average plateau depolarization was +6.4 ± 1.3 mV (see Figure 1A for an example, Figure 1B for a summary of data). The delay included the time required for diffusion of ETX in the extracellular medium and intrinsic effect of ETX. The effect on V_m_ of an application of 10^−4^ M BaCl_2_ (Figure 1D for a typical example), a treatment known to completely block all types of Kir currents, was also examined. In nine cells (from nine different culture dishes), the RMP was −53.1 ± 4.4 mV before and −0.5 ± 5.0 mV after BaCl_2_ application (*n* = 9, *p* < 0.001, paired Student’ t-test). The mean depolarization induced by Ba^2+^ ions was +54 ± 1.3 mV, and its delay was 20 ± 3 s (*n* = 9, significantly different from values obtained with ETX, *p* < 0.001, Mann Whitney Rank sum test). The depolarizing effect of ETX suggests that this toxin may act directly or indirectly on one or several types of the membrane channels controlling RMP in oligodendrocytes.

### 2.2. ETX Reduces Amplitude of Voltage-Dependent Kir Current in Cultured Rat Oligodendrocytes

Using the patch-clamp technique under the whole-cell clamp mode, we recorded the evoked membrane currents in rat oligodendrocyte. To establish the I_m_=*f*(V_m_) relationship under baseline conditions, oligodendrocytes membrane potential was held at −80 or −60 mV (depending on their initial RMP). Then, they were subjected to several series of twin symmetrical square changes in V_m_, increasing by step of 5 mV (see the Material and Methods for protocol and analysis; the experimental design is shown on top of Figure 2A). Typical examples of the corresponding evoked membrane currents (I_m_) are shown in Figure 2A (black traces). Amplitudes of the outward (i.e., positive) and inward (negative) currents were measured at their maximum (i.e., the peak detected after the transient capacitance artifact, as indicated by the black circles in Figure 2A_1_). Figure 2B (filled black circles) shows the mean relationship I_m_ = *f*(V_m_) established under baseline conditions. It was made by averaging (*n* = 12) of the mean individual cell I_m_ determined at each V_m_. Note that this I_m_ = *f*(V_m_) has not been corrected for the passive leak current. The inwardly rectifying shape of relationship I_m_ = *f*(V_m_) indicates the presence of a marked Kir current.

After membrane current recordings were made during the control period, ETX (10^−7^ M, final concentration in the extracellular medium) was applied (19 recorded oligodendrocytes from 19 different culture dishes). After 7 min of incubation in the presence of the toxin, the stimulation protocols were repeated again for a duration of 12–15 min. 16 of the 19 cells responded to the ETX with changes in the whole-cell currents. In 12 of them, the maximum amplitude of current evoked by hyperpolarizing stimuli was reduced (see typical current traces in Figure 2A_1_, red). In the four other responding oligodendrocytes, ETX effect manifested by a decrease in peak amplitude and slowing of membrane current inactivation leading to an increase in current plateau (Figure 2A_2_). In three of the 19 cells (16%), no effect on membrane current was detected (Figure 2A_3_). For each of these three cells, for each of the V_m_ applied to the cells, individual cell I_m_ ± SEM determined before and after ETX were not statistically different. Figure 2A_4_ summarizes the relative proportions of cells responding or not to ETX. Note that the proportion of cells with an unaffected whole-cell current is close to that of oligodendrocytes (three of 15, 20%) whose RMP has not been modified after ETX. Figure 2B,C summarize the decrease in amplitude of the inward membrane current determined 6 min after the application of ETX in the external medium. The mean I_m_ = *f*(V_m_) relationships, established before (black filled circles) and after ETX (red filled circles) (mean ± SEM, *n* = 12 different ETX-sensitive oligodendrocytes) indicates that this toxin blocks a significant fraction of the membrane current evoked by hyperpolarization (Wilcoxon matched-pairs signed-ranks test: *p* < 0.001; *n* = 12). Figure 2C shows the dispersion of peak current amplitudes recorded after a hyperpolarization to −130 mV, before and after application of ETX (10^−7^ M) to the bath for 6 ± 0.7 min (*n* = 12, same cells as in Figure 2B, *p* < 0.001, Wilcoxon Signed Rank Test). 

The membrane current fraction sensitive to ETX (I_ETX_) was isolated by subtracting the current recorded after toxin application to the current recorded before toxin application (see an example of the resulting current traces in Figure 2D). The 12 individual I_ETX_ = *f*(V_m_) relationships were pooled and averaged to establish the mean I_ETX_ = *f*(V_m_) relationship shown in Figure 2E. I_ETX_ displays the typical inwardly rectifying properties of Kir currents.

### 2.3. Further Characterization of the Fraction of Kir Current Inhibited by ETX

The next series of experiments were aimed at determining whether I_ETX_ is a Kir current. In a series of 5 experiments, oligodendrocytes whole-cell membrane currents were recorded (as described in §2.1) under 3 conditions: (a) control (Figure 3Aa), (b) then at least 6 min after addition of ETX (10^−7^ M, final) (Figure 3Ab), and (c) at least 1 min after further addition of 10^−4^ M BaCl_2_ (Figure 3Ac). The amplitudes of the evoked currents was determined at their maximum, after the capacitance artifact. The corresponding relationships I_m_ = *f*(V_m_) (mean ± SEM, *n* = 5) are shown in Figure 3C. The three I_m_ = *f*(V_m_) relationships were significantly distinct (Wilcoxon matched-pairs signed-ranks test: *p* < 0.001; *n* = 5).

In the oligodendrocyte membrane, Kir channels activity largely dominates over that of other voltage-dependent membrane channels (see Introduction). After the application of BaCl_2_, which blocks the Kir channels, only the passive membrane conductance remains. The relationship I_m_=*f*(V_m_) corresponding to the leak current is almost linear (Figure 3C, curve ‘c’, open triangles). The leak current contributes about 1/3 of the amplitude of the whole-cell current recorded under control conditions.

The voltage-dependent component of whole-cell current in oligodendrocytes (i.e., whole-cell current corrected for the passive leak current) was obtained by subtracting the current traces remaining after further addition of BaCl_2_ to the control ones (Figure 3D, black symbols, ‘a–c’ curve). This current displayed the strong rectifying properties of a Kir current. To confirm that it is indeed a Kir-type current, in a separate set of experiments, oligodendrocytes were subjected to 10^−4^ M BaCl_2_. After this treatment, only remained the leak current (data not shown) with a linear relationship I_m_ = *f*(V_m_) similar to that displayed in Figure 3C (curve ‘c’).

I_ETX_ (i.e., the fraction of the voltage-dependent current sensitive to ETX) was obtained (as described in § 2.2) by subtracting each of the traces of membrane current remaining after ETX from those recorded under control (Figure 3B, a–b; I_m_ = *f*(V_m_) curve in Figure 3D, red symbols, mean ± SEM, designated as ‘a–b’). I_ETX_ displayed the relationship I_m_ = *f*(V_m_) typical of a Kir current (Figure 3D, red symbols). When ETX was applied after BaCl_2_ has been added (i.e., when only the leak current remains), no change in the whole-cell membrane current was observed (data not shown). For instance, following the application of BaCl_2_, I_m_ = −27 ± 4 pA at V_m_ =−90 mV, and after subsequent addition of ETX (10^−7^ M, final) I_m_ = −39 ± 15, (mean ± SEM, *n* = 5, n.s.) at the same V_m_. The fact that the prior inhibition of Kir channels by BaCl_2_ occludes ETX effects provided support to propose that I_ETX_ is a fraction of I_Kir_.

Subtracting the current traces obtained after adding ETX and then BaCl_2_ (traces ‘c’ in Figure 3A) from those obtained after ETX (traces ‘b’ in Figure 3A) made it possible to establish I_m_=*f*(V_m_) curves for the voltage-dependent current fraction insensitive to ETX but sensitive to Ba^2+^ (Figure 3D, open symbols, ‘b–c’ curve).

The three I_m_=*f*(V_m_) curves in Figure 3C,D were significantly different from each over (Wilcoxon matched-pairs signed-ranks test: *p* < 0.001; *n* = 5) in terms of amplitude. However, when corrected for the leak current (Figure 3D) their voltage-dependence appeared similar. This was assessed in Figure 3F that shows that the three curves I_m_ = *f*(V_m_) obtained by subtraction of current traces and normalized to their respective maximum amplitude have almost similar voltage-dependence and rectifying properties. Importantly, there is no significant difference between the two components of I_Kir_: I_ETX_ and I_Ba^2+^ sensitive but ETX insensitive_ (*p* = 0.63, Wilcoxon signed-ranks test). The fraction of the evoked voltage-dependent membrane current (i.e., I_Kir_) inhibited by ETX was expressed (in %) by the ratio 100 × (‘a–b’) / (‘a–c’) using the data from Figure 3D (filled squares). This revealed that the extent inhibition of I_Kir_ by ETX was between 40 and 50% in this series of five experiments.

### 2.4. ETX Blocks a Large Conductance Kir Channel Type

We analyzed the effects of ETX on the unitary activity of Kir channels on intact oligodendrocytes, using the patch-clamp method under the cell-attached configuration. Steady-state hyperpolarization of the membrane patches revealed the presence of marked ionic channel activity (openings and closures), in the recorded patches. When membranes patches were maintained at a potential (V_patch_) of −40 mV, relative to the oligodendrocyte RMP (i.e., membrane patch more hyperpolarized by 40 mV than RMP) more, two different types of unitary channel currents were observed (Figure 4A,B, left panels). The unitary amplitude of the channel current was obtained by analyzing the recordings from membranes patches showing the activity of only a single channel. The mean channel amplitude (i_channel_) (right panels in A and B) was determined by adjusting the channel current amplitude histogram distribution by a Gaussian function. According to Ohms’ law, i_channel_ = γ_channel_ ∗ V_patch_. Hence, channel opening conductance was determined from the slopes of i_channel_=*f*(V_patch_) relationships, from -120 to +60 mV relative to oligodendrocyte RMP. Figure 4C summarizes data from 12 oligodendrocytes. Each sub-dataset was well adjusted using a simple linear relationship, which crossed the V_patch_ axis at ≈ +55 mV. Since the RMP in oligodendrocytes as been determined ≈ −55 mV (§ 2.1, Figure 1), when V_patch_ ≈ +55 mV, the corresponding V_m_ would be ≈ 0 mV. Under our experimental conditions, the concentration of K^+^ in the patch pipette was very high (see the Material and Methods) and should be in the same range as the intracellular concentration of K^+^. Thus, E_K_+ would be close to 0 mV, which corresponds to a V_patch_ ≈ +55 mV. The observed reversal potential for i_channel_ (i.e., i_channel_ = 0 pA at V_patch_ ≈ +55 mV, Figure 4C) indicates that the recorded channels are highly, if not only, selective for K^+^ ions, as expected from a Kir channel. The i_channel_ = *f*(V_patch_) slopes permitted determination of the opening elementary conductance of the two types of Kir channels, γ_channel_: 8.5 pS (small) and 25 pS (large), respectively.

The effect of ETX on Kir channels activity was examined. In the first series of experiments, the membrane patches were hyperpolarized by 40 mV relative to RMP for the whole duration of the experiments (Figure 5A). In the second series, patches were maintained at RMP except for 10 episodes of 500 ms duration where they were hyperpolarized by 40 mV from RMP (Figure 5B) to avoid a possible inactivation of the channel as a function of time. Whatever the used protocol, the result of an application of ETX was the same. In 9 of the 11 experiments, the addition of ETX (10^−7^ M, final) to the extracellular medium surrounding the patch pipette resulted in a rapid and marked reduction in Kir channel activity, with the large openings almost completely disappearing. In two of the 11 experiments, no ETX effects on Kir channel activity were detected. Figure 5A illustrates examples of channel activities recorded before, 42 and 72 s after ETX was applied to the bath (patch continuously maintained at −40 mV; the plots shown are from different recording times of the same patch). Figure 5B (top) illustrates 3 of 10 traces showing the activity of the high-conductance Kir channels, evoked by transient hyperpolarizing square steps (by 40 mV for 500 ms), before (left) and 2 min after (right) the application of ETX in the bath. Note that in the illustrated patch recordings (Figure 5B), only the activity of the high conductance channels was present during the control period. Moreover, the disappearance of the large openings is not associated with the appearance of small openings. The ensemble averages of single-channel patch currents obtained from 10 recordings made before and 2 min after ETX are shown at the bottom of Figure 5B. While, under control, large activities of opening are frequent enough to generate sustained evoked macroscopic Kir current, after ETX, this current was almost null. This suggests that the inhibition of I_Kir_ reported in Figure 2 and Figure 3 may be due to inhibition of such a type of large conductance channel.

In another series of experiments, we analyzed the activity of channels directly exposed to ETX in a long-lasting manner. When oligodendrocytes were incubated with ETX for at least 15 min before cell-attached patch recordings were made (*n* = 4 experiments), openings of the high-conductance channels were absent or very rare (0.001 s^−1^ in presence of ETX as compared to 1.5 s^−1^ before the toxin was applied). The low amplitude openings were still present (Figure 6), indicating that the low-conductance Kir channels are resistant to ETX, even after a long-term direct exposure to the toxin. The relationship i_channel_ = *f*(V_patch_) (determined as explained above) corresponding to these ETX-resistant Kir activities was similar to that established under control condition for small conductance Kir openings (compare Figure 6 with Figure 4C). Their conductance (γ_ETX-resistant channel_ = 9.8 pS) was close to that determined (γ_small-channel_ = 8.5 pS) under control conditions. 

## 3. Discussion

In this study, we report several observations revealing new ETX actions on rat oligodendrocytes in primary cerebellar cultures. (i) ETX induces small but significant RMP depolarization (by +6.4 mV); (ii) ETX reduces amplitude of evoked transmembrane current of type Kir; (iii) ETX induces inhibition of the activity of the high conductance Kir channel (~25 pS) without affecting that of the low conductance channels (<10 pS). In this discussion, we examine whether these different observations are caused by direct or indirect effects of ETX, are causally related to each other, and how this can inform ETX mechanisms.

### 3.1. Do the Observed Effects Result from a Direct Action of ETX on Oligodendrocytes?

The deleterious effect of ETX on oligodendrocytes leads to myelin damage due to toxin action against oligodendrocytes [9,12,13,14] as well as inhibition of a Kir channel type (this study). The first question to be asked is whether the ETX effects reported in this article result from the direct action of the toxin on rat oligodendrocytes or are indirect, i.e., mediated by the attack of another cell type present in the primary culture. ETX does not bind to astrocytes (mouse, rat) and binds with a high affinity to oligodendrocytes [8,9,12]. Unlike in mice [8], ETX does not bind to rat neurons [9]. In addition, the primary culture protocol used in this study does not allow neurons to survive [9]. Therefore, all toxin effects reported here result only from a direct action of EXT on rat oligodendrocytes.

### 3.2. Relationship between Membrane Depolarization and Kir Channel Inhibition

ETX can act on rat oligodendrocytes without forming pores in their plasma membrane [9]. This study fully confirms this deduction. The amplitude of the whole-cell current is decreased after ETX (see the relationships I_m_=*f*(V_m_) in Figure 2 and Figure 3). Such a result cannot be due to membrane permeabilization. Indeed, any decrease in membrane resistance should result in a dramatic increase in whole-cell membrane current due to an increase of passive leak current, as we have already reported in the case of the cerebellar granule cells [8] and renal cells [23]. Therefore, the inhibition of Kir channel activity reported in this article is not related to the ETX pore-forming activity documented in other preparations [1,8,23].

The intracellular block of open Kir channels by Mg^2+^ and polyamines increases significantly with depolarization of the plasma membrane [18], which explains the typical inward-rectification of the curve I_m_ = *f*(V_m_) typical of the Kir current (examples are given in Figure 2 and Figure 3). Moreover, the various types of Kir channels do not show the same degree of inward rectification. Some of them are strong rectifiers (such as Kir2 channels) while others are less rectifiers (such as the Kir4 family channels) [18]. The question arises as to whether the specific inhibition of large amplitude Kir openings after ETX is a cause or a consequence of the small depolarization (+6.4 mV) of oligodendrocytes that is induced by the toxin. The membrane-current fraction inhibited by ETX, I_ETX_, has the same characteristics as the toxin-resistant Kir current fraction: inward-rectification (Figure 3D,E) and sensitivity to Ba^2+^ ions (total occlusion of the ETX effect by pre-incubation of oligodendrocytes in the presence of Ba^2+^ ions, see §2.3). Figure 3D (‘a–c’ curve) shows that to obtain a one-third inhibition of I_Kir_ it would require a depolarization of V_m_ by at least 30 mV. This excludes the possibility that the small membrane depolarization induced by ETX (+6.4 mV) may be responsible for the marked inhibition of the Kir current (Figure 3F). Conversely, in line with the well-established role of Kir channels in maintaining the RMP [17,18,19,20], inhibition of the Kir current could be the cause of the +6.4 mV RMP depolarization induced by ETX. To test this possibility, RMP (i.e., the membrane potential, E_m_) was modeled using the Goldman–Hodgkin–Katz constant field equation [24]. We used the extracellular and intra-pipette concentrations described in the Material and Methods section. The weight of the different relative permeabilities must be consistent with the notion that at rest, *P*_Kir_ is much higher than the other ones in oligodendrocytes [17,21,22]. When adjusting the relative membrane permeabilities (*P*) for ions (*P*_Kir_ = 0.865; *P*_non-Kir_ = 0.045; *P*_Na^+^_ = 0.045 and *P*_Cl_^−^ = 0.045), this yields an E_m_ ~−55 mV (as observed control RMP, Figure 1B). E_m_ ~0 mV when *P*_Kir_ becomes zero (as shown in the presence of 10^−4^ M BaCl_2__,_
Figure 1D). Using this model, a 29% *P*_Kir_ reduction results in a depolarization of ~6.4 mV, as that observed after ETX (Figure 1). Thus, the reduction of about one-third of I_Kir_ after ETX would be sufficient to induce the small depolarization of oligodendrocytes that we observed after the application of the toxin. 

### 3.3. What Might Be the Kir Channel Type Inhibited by ETX?

Under the primary culture conditions used, at least two types of Kir channels are active in the oligodendrocyte membrane (Figure 4 and Figure 5). Their unit conductance is 25 pS and <10 pS, respectively. This is consistent with the fact that mature (i.e., myelinating) and immature oligodendrocytes express several types and subtypes of Kir channels [17]. The most abundant type in oligodendrocytes is Kir4.1 (intermediate rectifier [18]), which is specific for glia in the central nervous system [17,25,26]. The other Kir expressed in oligodendrocytes are Kir5.1 [17,25], Kir2.1 and Kir2.3 (which are strong rectifiers) [18]. The conductance of Kir channels varies with subtypes (homomeric Kir2.1: ~20–40 pS, Kir2.3: ~13–14 pS, Kir4.1: ~20–40 pS; Kir4.1/Kir5.1: ~40–60 pS) [18]. Since the rectifying properties of ETX-sensitive and -insensitive Kir are very similar (Figure 3E) it is difficult to propose an unambiguous identity for the Kir channels of small and large conductance that we have observed. Given its low conductance, the 8.5–10 pS Kir channel (i.e., the ETX insensitive channel) could correspond to the homomeric Kir2.3 channel (a strong rectifier [18]). The high conductance Kir channel (25 pS) (i.e., the one inhibited by ETX) could correspond to the Kir 2.1 (strong rectifier) or homomeric Kir4.1 (intermediate rectifier).

### 3.4. ETX Inhibits Large Amplitude Unitary Kir Channel Activity via An Intracellular Mechanism

The application of ETX in the extracellular medium during cell recording inhibits the 25 pS Kir channels enclosed at the tip of the patch pipette. The very high electric resistance (>Giga–Ohms) of the tight seal between the tip of the patch-pipette and the plasma membrane does not allow the diffusion of the bath ETX into the intra-pipette medium, thus preventing any direct action of the toxin on the Kir channels whose activity is recorded by the patch-pipette. To date, ETX has never been found internalized [23] or acting in the intracellular cytosolic compartment. In addition, the time taken by the ETX to depolarize the RMP is longer (by 33 s) than the time needed to observe a depolarization-induced by Ba^2+^ ions (§ 2.1) suggesting that the ETX delay is due to the mechanisms underlying its cellular effects rather than its diffusion in the extracellular medium after its application to the bath. Since different intracellular pathways can modulate Kir channel activity [18], the most likely interpretation of our findings is that following binding of ETX to its extracellular membrane receptor, there is an activation of an intracellular signaling resulting in the indirect inhibition of large conductance Kir channels including those enclosed in the recorded membrane patch. The signaling pathway involved is undefined and deserves further investigations. For example, Kir4.1 channel activity depends on the concentration of intracellular ATP [17,18] and ETX is known to cause ATP depletion in certain cell types such as renal epithelial cells [23,27] and epithelial thyroid cells [28]. Since the MAL protein identified as a potential receptor for ETX on oligodendrocytes [11] has not been reported activating intracellular signaling pathways, it is possible that the observed effects result from binding of ETX to another type of receptor.

### 3.5. Heterogeneity of ETX Effects on Oligodendrocytes

In this study, we found that ETX had no effect on a fraction of the recorded oligodendrocytes. Indeed, (i) the RMP remains unchanged after ETX in three of the 15 (20%) cells (§ 2.1), (ii) three of the 19 (16%) cells showed unchanged whole-cell current after ETX application (§ 2.2), (iii) and in two of the 11 (18%) membrane patches no change in Kir channel activity was detected after ETX (§ 2.4). Using the same type of primary culture and experimental conditions as in this study, we reported the absence of ETX binding on 21% of cultured oligodendrocytes [9]. The lack of ETX effect observed in this study (16–20% of cells) could, therefore, be due to the lack of ETX binding to some of oligodendrocytes. Other sources of variability deserve to be considered. We found the presence of at least two types of Kir channels in the oligodendrocytes, one of which is not inhibited by ETX. Thus, resistance to the action of ETX could also result from the preferential expression by certain oligodendrocytes of ETX-insensitive channels. It should be noted that oligodendrocytes express a number of types of Kir channels (see §3.4) but the types expressed vary according to conditions and stage of development [17,18,25]. 

### 3.6. Possible Contribution of Kir Inihibition by ETX to Myelin Dysfunction

Kir channels play an essential role in the homeostasis of extracellular K^+^ in the white matter. This allows maintaining axonal RMP and nodal function, and favors rapid propagation of action potentials along axons [17,18,19,20]. The genetic deletion of the Kir4.1 channel in mature mouse oligodendrocytes does not kill these cells but alters myelin. This results in spontaneous seizures and abnormal motor behavior [29]. These manifestations resemble some of those displayed when animals are poisoned by ETX [3,4] or after ETX is injected into the hippocampus [30]. Whether these effects result directly from inhibition of a Kir-subtype or demyelination needs further investigations. The deletion of Kir4.1, which regulates the clearance of extracellular K^+^ in the white matter; prevents axonal transmission of high-frequency repetitive action potentials [29]. Therefore, we speculate that the inhibition in oligodendrocytes of one of the types of Kir channels by ETX could explain why high-frequency nerve conduction is strongly altered after ETX [14]. Given the high dependence of the activity of the membrane amino-acid transporter to V_m_, the modification of Kir channel activity and the depolarized RMP that ensues results in an alteration of the glutamate transporter activity [31,32]. Therefore, inhibition of Kir current by ETX might be the cause of the extracellular increase in glutamate concentration that we have observed when oligodendrocytes have been incubated with ETX [9]. Since the demyelination induced by ETX in cerebellum slices needs activation of the glutamates receptors mGluR1 and NMDA-R [9], we propose that the inhibition of one type of Kir channel in oligodendrocytes might be the initial event triggering demyelination process.

## 4. Conclusions

This study revealed the new effects of ETX on oligodendrocytes. They consist of decreased amplitude of membrane Kir current due to selective inhibition of activity of high-conductance Kir channels (25 pS) and ensuing small membrane depolarization. The inhibition of high-conductance Kir channels does not result from a direct action of ETX on them, and involves undefined intracellular mechanisms. Inhibition of Kir channels by ETX could represent the first step in a cascade of events (decrease in I_Kir_ current and RMP depolarization—this article—increase in extracellular glutamate, activation of mGluR1, and activation of intracellular Ca^2+^ signaling [9]) causing myelin damage and induction of the demyelination process [9,10,12,13].

## 5. Experimental Procedures

### 5.1. Ethics Statement

All experimental procedures were in accordance with European and French guidelines for animal experimentation, and have been approved by the Bas-Rhin veterinary office, Strasbourg, France (authorization number 67–26 to BP) and by the Paris veterinary office, France (authorization number 75–279 to MP) applying at the time when experiments were performed.

### 5.2. Epsilon Toxin

ETX was purified from *C. perfringens* type D strain NCTC2062 as previously described [33]. Aliquots containing 10 µL of ETX 10^−5^ M diluted in physiological medium (§ 5.4) were prepared and stored frozen. When needed, 1 aliquot (10 µL) was added to the extracellular medium (1 mL) bathing the oligodendrocytes in primary cultures. Extracellular medium was gently mixed using an automatic pipette (200 µL tip) to obtain a final concentration of 10^−7^ M.

### 5.3. Cerebellar Primary Cultures Containing Oligodendrocytes and Astrocytes

Primary cultures of newborn (P0–P1) rat cerebellum (Wistar rats from the animal facility of CNRS and University, UMS3415 Chronobiotron, Strasbourg) have been prepared as detailed elsewhere [9]. In summary, culture conditions allow the selection of oligodendrocytes and astrocytes while eliminating all neurons. Freshly dissociated rat cerebellar cells were re-suspended in DMEM supplemented with high glucose, 5% decomplemented horse serum and 5% decomplemented fetal calf serum. After two days of culture in this medium, the cells were rinsed and then exposed for at least four days to a chemically defined serum-free medium composed of DMEM supplemented only with high glucose. Oligodendrocyte-based experiments were performed at least 4 days after transfer to this second stage of culture.

### 5.4. Electrophysiological Recordings

Before the electrophysiological experiments, the culture medium was replaced by an external solution containing in mM: NaCl 130, KCl 2.6, CaCl_2_ 2, MgCl_2_ 2, HEPES 10, pH 7.4 with Tris/OH. Mature oligodendrocytes were selected, by eye, according to their typical round cell body shape from which processes emerge radially. Electrophysiological recordings of Vm or whole-cell current (I_m_) were made using the patch-clamp technique in whole-cell configuration, in the current- or voltage-clamp mode. Recording of unitary activity of the membrane ionic channels was made using the cell-attached configuration under the voltage-clamp mode. Recording micropipettes (resistance 5 ΩM) were filled with a solution containing in mM: potassium glutamate (KGlu) or KCl 130, NaCl 2, MgCl_2_ 1, CaCl_2_ 2, EGTA 10, HEPES 10, the pH 7.2. The correction for the liquid junction potential was made a priori. The V_m_ of each recorded cell was held at −60 or −80mV under voltage-clamp. 

To establish the relationship between the amplitude of I_m_ and V_m_ (I_m_=*f*(V_m_)), each of the recorded cells were submitted to series of twin symmetrical square changes in V_m_, 400 ms duration each, every 1 s, increasing by step of 5 mV (see experimental design in the top of Figure 2A). For each cell, this protocol was repeated several times before and after treatment.

The electrophysiological voltage or current signals, under the whole-cell or cell-attached configuration, were acquired (filtered at 5 kHz) using an Axopatch 200A amplifier (axon Instrument) and digitized at 50 kHz using a digidata-1322A (Axon Instruments) before being recorded using the Clampex-10.2 software. Clampfit-10.2 software was used for offline analysis. Before the analysis, all king of the digitized signals were re-filtered with a cut-off frequency of 1 kHz (whole cell current traces) or 1.5 kHz (unitary channel activity) [34,35]. The mean I_m_ amplitudes determined at the same V_m_ with a given cell were averaged. Then, they were pooled with those obtained from the other cells at the same V_m_, and averaged to provide mean current (I_m_) amplitude ± SEM at each V_m_. This allowed to establish the relationship I_m_ = *f*(V_m_) for the pooled cells under each of the different experimental conditions. The statistical significance of the differences in the mean I_m_ = *f*(V_m_) curves established before and after a treatment was tested using the Wilcoxon matched-pairs signed-ranks test.

The analysis of the unitary activity of ion channels was carried out as described elsewhere [34,35]. The detection threshold for opening and closing transitions has been set at 50% of the open level of each type of event analyzed. All events with a duration not exceeding 0.5 ms were rejected. The average amplitude of the unitary currents was obtained by adjusting the amplitude histogram with a Gaussian distribution using Clampfit 10.2 (Axon Instruments) software. The average amplitude of the membrane noise was similarly determined from the analysis of recording episodes that had no channel opening. During the analysis of the conductance of the channels, the normality of their distribution was assessed using the Shapiro-Wilk test and equal variance test before running an unpaired Student’*t*-test.

### 5.5. Graphs, Fitting Procedures and Statistical Analysis

The statistical analysis and graph presentations of the data were made using Sigma-plot-12.5 software. The data obtained before treatment with ETX or BaCl_2_, i.e., during the control period of time, were designed as ‘control’ or ‘baseline’ in the text and figures. Average data are presented as mean ± SEM (otherwise stated). 

When cell-attached recordings are made, the actual membrane potential of the membrane patch (V_patch_) is a function of RMP: V_patch_ = RMP-V_pipette_. However, the actual RMP remains unknown. It is therefore specified in the text by how much (in mV) V_patch_ differs (hyperpolarized or depolarized) from RMP by mentioning the value of V_patch_ relative to RMP.

Statistical comparison of normally distributed data before and after treatment were tested using appropriate paired Student’t-test. When measurements were non-normally distributed, paired comparisons before and after were tested using the Wilcoxon Signed Rank Test, and unpaired comparisons using the Mann Whitney rank sum test. The data were considered statistically significant when probability *p* < 0.05; * denotes *p* between 0.05 and 0.01; **, *p* between 0.01 and 0.001; ***, *p* values < 0.001. Goodness of fit was determined based on the lower prediction error.

## Figures and Tables

**Figure 1 toxins-12-00036-f001:**
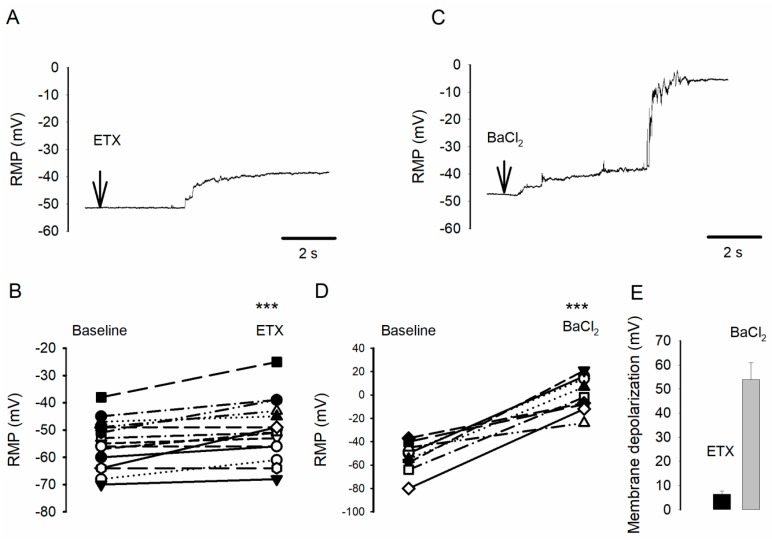
Epsilon toxin (ETX) induces oligodendrocyte depolarization. (**A**) Typical recording of the resting potential of the membrane (RMP) (under current clamp) of an oligodendrocyte, before and after ETX (10^−7^ M, final) has been applied in the surrounding medium, at the time indicated by an arrow. (**B**) Summary of *n* = 15 experiments (15 oligodendrocytes from 15 different culture dishes), during which the RMP was measured before (Baseline) and after application of ETX. (**C**,**D**) Same presentation as in A and B, except that BaCl_2_ (10^−4^ M, final) was used instead of ETX. (**E**) Mean amplitude ± SEM (*n* = 9, from nine culture dishes) of membrane depolarization (starting from baseline RMP) induced by ETX (black bar) or Ba^2+^ ions (grey bar). See the text for statistical significance.

**Figure 2 toxins-12-00036-f002:**
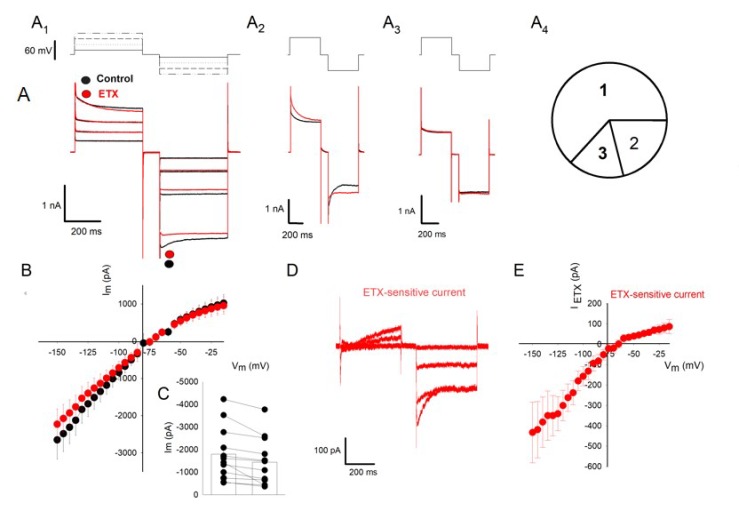
ETX inhibits a fraction of a transmembrane inward rectifying current in oligodendrocytes. (**A**) The currents traces (whole-cell clamp configuration) evoked by successive pairs of depolarizing and hyperpolarizing steps, from a holding potential of −80 mV, before (black traces) and after application of 10^−7^ M ETX (red traces). The same protocol was repeated by increasing the stimulation amplitude by 5 mV steps. The experimental paradigm is illustrated at the top of panels A_1_ to A_3_. A_1_: traces of current of an oligodendrocyte responding to ETX by a decrease in the whole-cell membrane current. Only four traces are illustrated. A_2_: traces of current of an oligodendrocyte responding to ETX by a decrease in peak current and an increase in plateau current. A_3_: example of a current trace unchanged following ETX application. A_4_: Pie chart showing the respective proportion of 19 recorded cells responding respectively with a decreased whole-cell current (1; *n* = 12; see A_1_), a modified kinetics of whole-cell current (2; *n* = 4; see A_2_), and no change (3, *n* = 3; see A_3_). (**B**) The mean relationship (± SEM) I_m_ = *f*(V_m_) -not corrected for membrane leak current- established by pooling data from the subset of the 12 oligodendrocytes responding to ETX with a decreased whole-cell current (traces in A_1_) (12 different culture dishes). Depending on the initial RMP, holding potential has been set to −80 mV or −60 mV. I_m_ was measured on each trace at peak current (after the capacitance artifact) as indicated in panel A_1_ by filled circles before (black) and after ETX (red). (**C**) I_m_ measured at V_m_ = −130 mV, before (left) and after ETX (right) in the same 12 cells as in B. The histogram bars in the scatter plots represent mean I_m_ (± SEM). (**D**) Example of ETX-sensitive current obtained by subtracting the traces recorded after 7 min of incubation with the toxin from those recorded before (same cell as in A_1_). (**E**) The relationship I_ETX_ = *f*(V_m_) obtained by pooling the maximum amplitudes of subtracting membrane current traces (before/after ETX) (mean ± SEM) determined in the ETX-sensitive oligodendrocytes (same 12 cells as in B).

**Figure 3 toxins-12-00036-f003:**
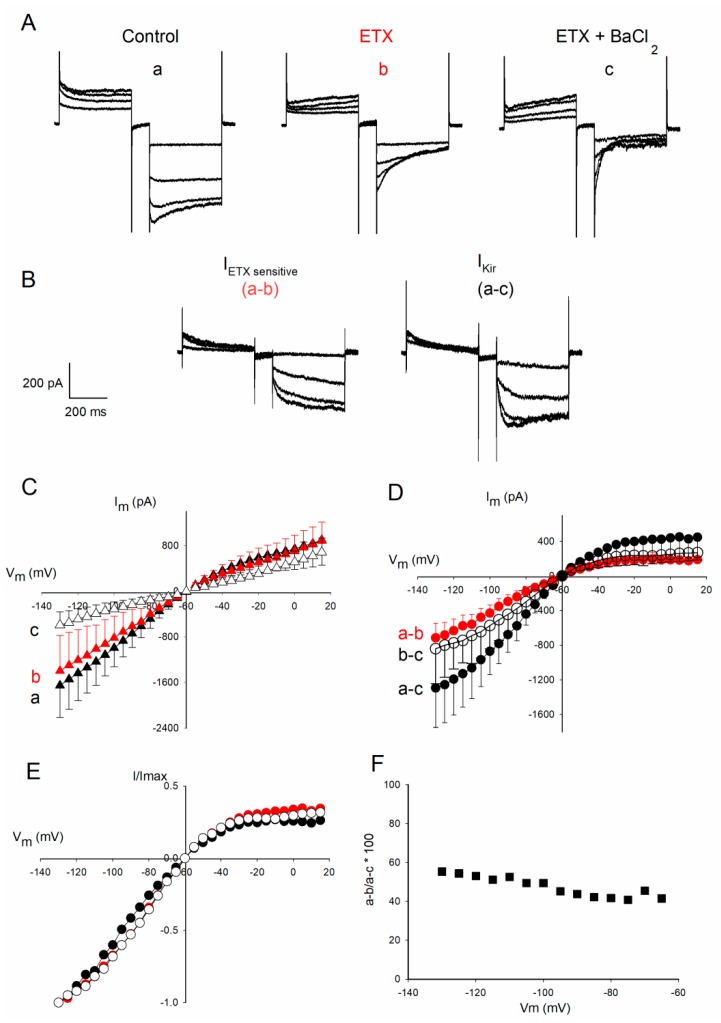
The ETX sensitive membrane current has the characteristics of the Kir current. (**A**) Examples of traces of current of the same oligodendrocyte recorded before (a), at least 5 min after 10^−7^ M ETX (b), and at least 3 min after an additional application of 10^−4^ M BaCl_2_ (c). (**B**) The corresponding subtraction current traces: ETX-sensitive current traces were obtained by subtracting ‘b traces’ from ‘a traces’ (noted as ‘a–b’). The whole-cell current corrected for the leak current was obtained by subtracting the traces ‘c’ from those of ‘a’; they are called ‘a–c’. (**C**) The relationships I_m_=*f*(V_m_) corresponding to the three conditions mentioned in Aa, Ab, Ac (mean ± SEM, *n* = 5 oligodendrocytes from 5 different cultures). (**D**) The relationships I_m_ = *f*(V_m_) established from the subtractions ‘a–b’ (red symbols, i. e. I_ETX_), ‘a–c’ (filled black symbols, i. e. the whole current corrected for the leak current), and ‘b–c’ (open circles, i. e. the fraction of current insensitive to ETX and corrected for the leak current). (**E**) Same relations I_m_ = *f*(V_m_) as in panel D but after normalization to their respective maximum (same symbols and colors as in D). (**F**) Percentage of inhibition by ETX of the whole-cell current corrected for the leak current as a function of V_m_ determined from the data in panel D using the equation % inhibition = 100 × (a–b) / (a–c).

**Figure 4 toxins-12-00036-f004:**
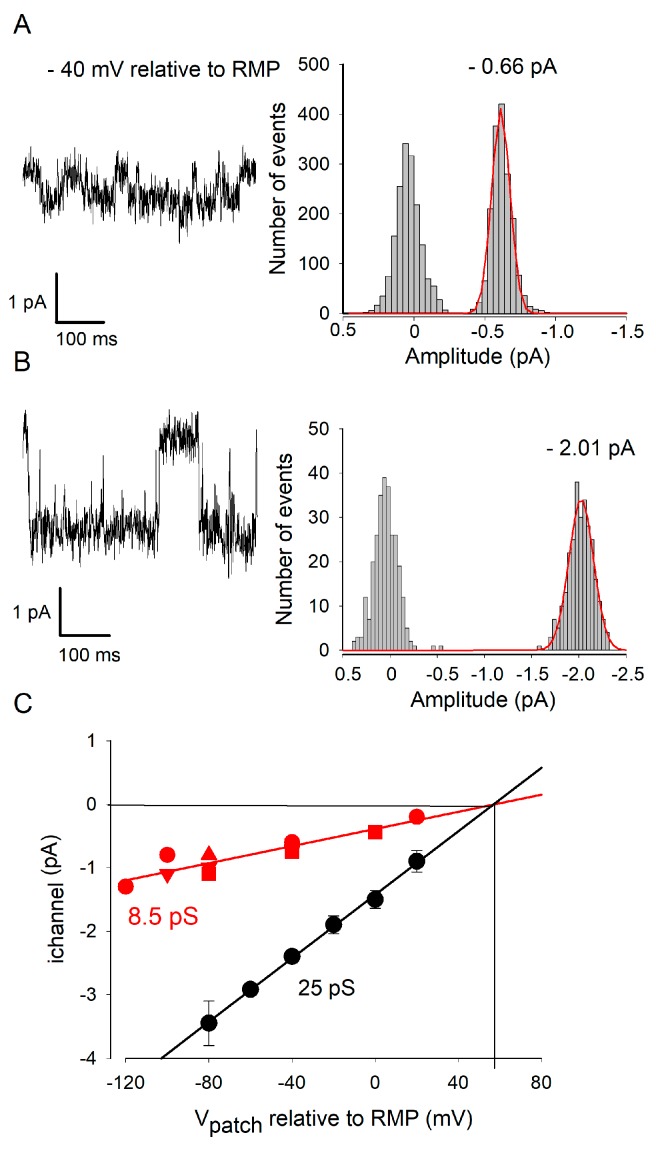
Two types of Kir channels are active in oligodendrocytes in culture. Single-channel activity was recorded on the oligodendrocyte membrane using the cell-attached configuration and with high intrapipette concentrations of K^+^. To evoke activity of the Kir channel, V_patch_ was imposed at −40 mV relative to RMP. (**A** and **B**) On the left, the plots illustrate examples of the two types of Kir channel openings (sudden downward current changes) found on the oligodendrocytes in culture. The panels on the right of A and B illustrate the amplitude distribution histograms of the current amplitudes of the channel currents (i_channel_). The data shown in A and B are from separate membrane patches. Note that the histogram mode on the left (near a current amplitude of 0 pA) correspond to fluctuations in membrane noise in the absence of a channel opening, while those on the right correspond to the amplitudes of unitary currents due to the opening of the channels (superimposed on fluctuations due to membrane noise). Each amplitude distribution histogram was best adjusted using a simple Gaussian function (indicated by a solid red line). This unimodal distribution indicates the presence of a single channel opening in the analyzed patch. The corresponding average i_channel_ amplitudes are indicated on the histograms. (**C**) The relationships i_channel_ = *f*(V_patch_) established for both types of channels (black: large amplitude channels, red: small amplitude channels). The average i_channel_ amplitude (mean ± SD) was established from pooling measurements obtained from analyzing seven (filled black circles) and five (filled red symbols) patches showing the activity of a single channel but characterized by a large or small unitary current at the same V_m_. Note that the fluctuation bars may be smaller than the size of the symbols. The data were better adjusted using a linear regression (solid straight line), whose slope provides a conductance of 25 pS (black line) and 8.5 pS (red line) for the analyzed channels, respectively.

**Figure 5 toxins-12-00036-f005:**
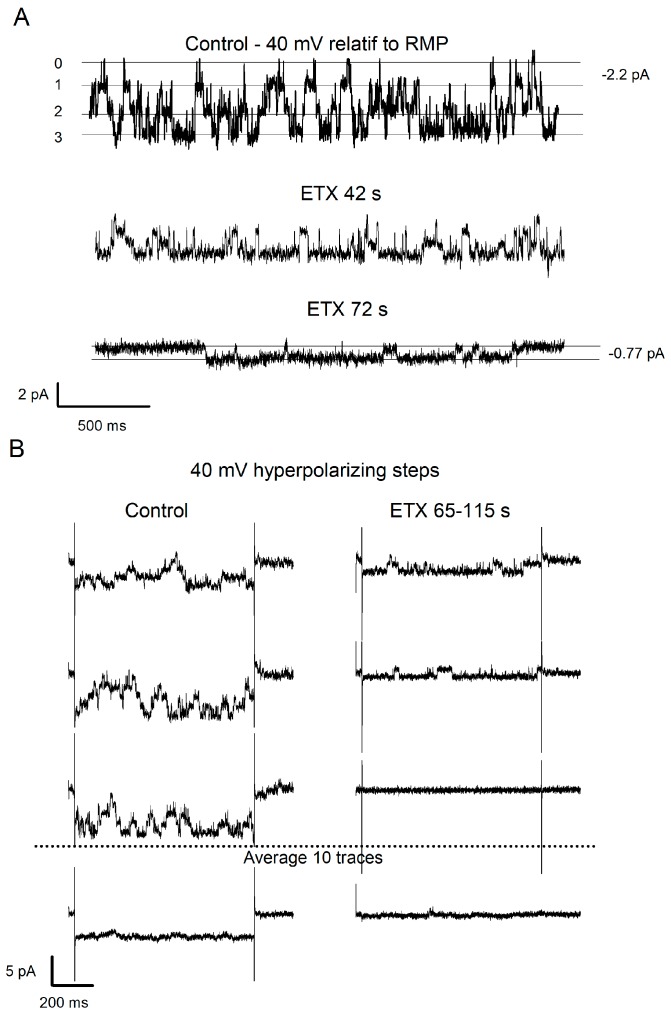
ETX abolished the activity of the large conductance Kir channel type. (**A**) Current traces showing channel opening activities recorded in a membrane patch containing several large and small conductance channels. V_patch_ was maintained permanently hyperpolarized at −40 mV compared to RMP. Note that before the application of ETX, this membrane patch (upper trace) shows the simultaneous activity of at least three channels with large amplitude unitary currents (indicated by three amplitude levels, increasing by 2.2 pA). Smaller unitary currents indicate the activity of low-conductance channels in this same membrane patch. The application of ETX (10^−7^ M) induces a reduction in the opening activity of channels characterized by large unitary currents (median and lower traces, same membrane patch as above). Note that 72 s after the application of ETX, only the activity of low-conductance channels (0.77 pA) (lower trace) remains. (**B**) Right and left, the three upper current traces (3 out of 10) come from the same membrane patch subjected to hyperpolarization steps for 500 ms at −40 mV (with respect to the RMP), before (left column) and after application of the ETX (10^−7^ M final) (right column). Note that in this membrane patch, only the opening activities of the high conductance channels were detected. In both columns, the bottom plot represents the average of 10 successive current traces of the same membrane patch, before or 2 min after the ETX application. Note that due to the lack of channel opening activity, the average amplitude current (right) is almost zero after ETX compared to that observed before the toxin application.

**Figure 6 toxins-12-00036-f006:**
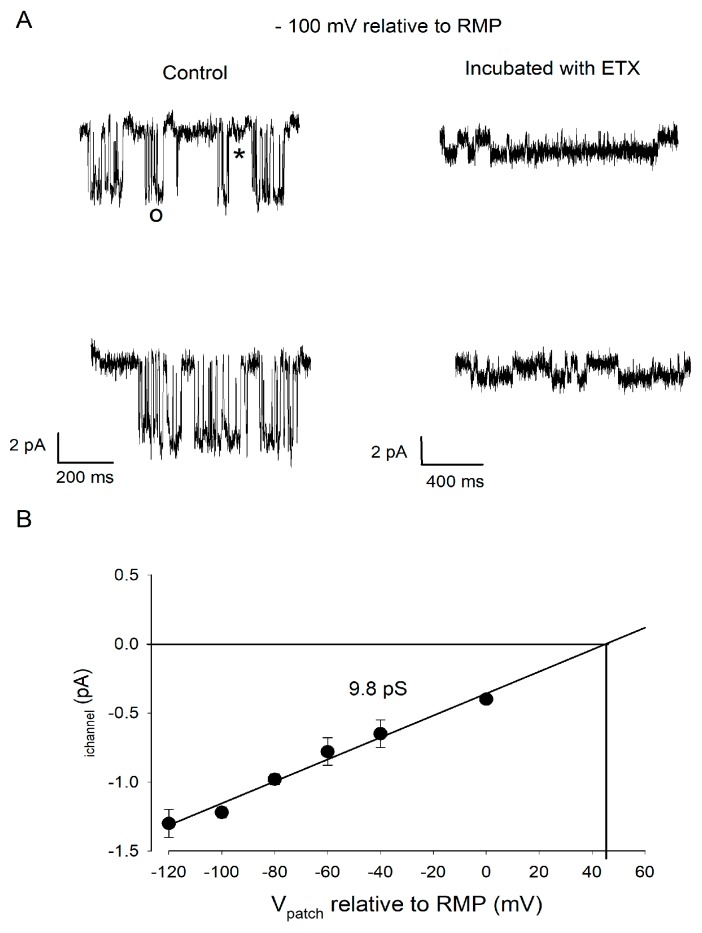
ETX-insensitive channel has a low conductance. (**A**) The traces on the left illustrate two examples of recordings of channel activity when V_patch_ was hyperpolarized by 100 mV relative to the RMP, before (left) and after (right) application of 10^−7^ M ETX for 15 min. Note the presence of large (o) and small (*) amplitude openings before toxin application (left). After ETX, only small amplitude openings were detected (right) indicating these Kir channels are insensitive to ETX. (**B**) Same presentation as in Figure 4C. The i_channel_ = *f*(V_patch_) relationship was established from the pooling of data from *n* = 4 experiments. Linear regression of the data (black line), the slope of which indicated that the Kir channels insensitive to ETX have a conductance of 9.8 pS.

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
