# Peer review of "Epsilon Toxin from Clostridium perfringens Causes Inhibition of Potassium inward Rectifier (Kir) Channels in Oligodendrocytes"

_toxins, 2020, doi:10.3390/toxins12010036_

Round 1
Reviewer 1 Report
In this study, the authors examined the effect of the Epsilon toxin (ETX) from C. perfringens on inhibiting the potassium inward rectifier channels in oligodendrocytes. The rationale of the study is that past studies have demonstrated that ETX can induce demyelination and alter nerve conduction, which are two early symptoms of mutiple sclerosis. In this study, the author demonstrated that ETX was able to induce small amount of transmembrane depolarozation and inhibition of the Kir transmembrane current. Furthermore, ETX was found to inhibit one of the Kir channels. Experimental design was sound and conclusion was reasonable. Syntax errors throughout need to be revised.
Reviewer 2 Report
In this study, the effect of the Epsilon toxin from Clostridium perfringens (ETX) on oligodendrocyte membrane currents was investigated. It was found that at a concentration of 100 nM, the toxin induced a small inhibition of membrane potassium currents under whole-cell voltage-clamp, and a small depolarizing shift of the membrane potential. When single-channel activity was measured, the same concentration of ETX abolished the current activity almost completely. Single-channel experiments in which the toxin was applied 15 min before seal formation suggested that of the two unitary conductances present in the patches, only the type with the higher conductance was inhibited.
Major points
Small effect of ETX at the concentration used, and inconsistency between experiments. In the first series of experiments (Figs. 1 and 2) it is shown that ETX at 100 nM induces a depolarizing shift of the resting membrane potential of +6.4 mV. Fig. 2 then shows that out of 19 experiments, ETX had no detectable effect on the current in 3 cells and did only affect the kinetics, but not the current amplitude in 4 other cells. In the remaining 12 cells, ETX inhibited the inward currents. The percentage of current inhibition is not discussed, but it was at most 20%. Barium inhibited this current almost completely (as shown in Fig. 2), indicating that the measured current is to at least 95% "Kir" current. Thus, the conclusion in Fig. 2 is that ETX did not inhibit the membrane current in 37% of the measured cells, and induced an inhibition of at most 20% of the Kir current in the remaining 63% of the cells. In Fig. 3, based on a dataset of 5 experiments, it was then concluded that the Kir current was inhibited with the same concentration of ETX by 38-61%. This finding is clearly not consistent with the observations in Fig. 2. In the single-channel experiments it appears that in the example of Fig. 5, ETX blocked the current completely. Single-channel measurements are for sure a powerful approach, but they are also risky, since one tries to conclude from the behavior of a very small number of channels on the behavior of a whole population of channels. The obvious test for the relevance of findings with single-channel experiments is the comparison of their average behavior to the whole-cell results. Here, the current inhibition by ETX in the single-channel experiments is much stronger than in the whole-cell experiments, which puts their relevance into question. In the context of the small effects of ETX in whole-cell experiments it would be important to know the concentration dependence of the established effects of ETX, and how this relates to the concentration dependence of Kir current inhibition.
Some practical/technical aspects of the experiments are not sufficiently described. The missing information needs to be provided in the manuscript. This includes the following points: a) How fast was the solution exchange, what was the speed of the perfusion, or in case the toxin was applied as a stock solution into the bath, indicate the concentration and composition of the stock solution, the volume of the chamber, and the time of diffusion. In the second case, the measurement of the "delay"(line 140 ff), would largely reflect the diffusion of the stock solution, and should be described as such. b) line 140ff, how was the delay defined? It may be more appropriate to use a rise time or Tau. c) it is stated on p.5 that the "peak" currents were measured as the current amplitude (average?) of the first 50ms of the trace. This seems very imprecise, and would still include the transients. Would it not be possible to measure at the real peak? d) Indicate after which time of incubation with ETX the measurements were made. In Fig. 2D it is indicated that this was 7 min; the incubation time was however not indicated for the other figures, and in the text (line 205) it is said that the incubation time was 6 min. e) for the single-channel experiments, clarify what "Vpatch relative to RMP (mV)" means, thus whether this is the voltage applied to the recording pipette or whether it is inversed, and re-check the discussion of the membrane potential on top of Fig. 8. For the single-channel experiments, indicate also if for the analysis an additional filtering (to the 5kHz filter) was used.
Specific points
Introduction, line 25, the indication (400'000 mouse LD100/mg protein) is not clear, please clarify. Lines 35-38, immunoreactivity in MS to ETX, indicating the numbers for the patients makes only sense if numbers for healthy controls are also provided. Origin of inward rectification (lines 56-58, and at other places). The inward rectification is due to intracellular block by magnesium or polyamines. Therefore, the statement that the inward rectification is due to changed open probability is, if not wrong, at least strongly misleading. Line 90, "to record membrane ionic..", you mean "to record unitary..." ? line 197 and 202/203, "no detectable effect", define the threshold value used to classify an amplitude as not detectable. In a few instances, conclusions are made that are not based on any statistical analysis. They should be backed by appropriate analyses. This involves (line 233/234) the conclusion that in Fig. 3C inhibition is greater at -130 than at -80 mV, and (lines 238-240) the conclusion that the voltage dependencies in Fig. 3D are different from each other. Fig. 5. It would be good to show a continuous trace of channel activity, with indication of the moment of solution change. Also, a quantitative summary of the inhibition of Kir currents in different patches should be provided. Fig. 6. Can the authors exclude that the small amplitude openings are not high conductance Kir channels that are partially blocked? The discussion could be shortened. The paragraph 4.5 is mostly speculation. Facts and hypotheses should be better distinguished.
Reviewer 3 Report
These are very interesting and novel data. However, several important questions should be addressed prior to the acceptance.

Round 2
Reviewer 3 Report
I went through the revised paper and I think it is acceptable now.